# Performance evaluation of inclusive green growth in China: Dynamic evolution, regional differences, and spatial correlation

**Yingchao Xu[1], Lu Li[2]\*, Shujian Xiang[3]**

1 School of Economics, Jiaxing University, Jiaxing, Zhejiang, China, 2 College of Data Science, Jiaxing University, Jiaxing, Zhejiang, China, 3 School of Statistics and Mathematics, Zhejiang Gongshang University, Hangzhou, Zhejiang, China

\* yulu0603@zjxu.edu.cn

**Data Availability Statement:** All relevant data are within the manuscript and its Supporting information.

**Funding:** This work was supported by grants from the National Social Science Fund of China

## Abstract

Inclusive green growth is an essential way to achieve sustainable development. We construct an index system for inclusive green growth performance levels (IGGPLs) in Chinese cities and measure the IGGPLs of 271 cities in the Chinese mainland from 2006 to 2020 based on the vertical and horizontal scatter degree method. We employ the Kernel density method, Dagum Gini coefficient method, Moran index method, and Markov chain method to investigate distribution evolution, regional disparities, spatial correlations, and state transition of IGGPLs at the city level, respectively. The research results reveal that: (1) China's IGGPL has improved rapidly, but regions with lower IGGPLs still predominate. The eastern region is far ahead, followed by the northeastern region, with the western and center regions trailing; (2) The development trends of IGGPLs in the eastern and central regions are positive, with no signs of polarization. Although polarizations are obvious in the western and northeastern but have been improved significantly; (3) Regional differences exist, but are gradually narrowing over time. By decomposing regional differences, we find that regional differences are the main cause of total differences; (4) The IGGPLs of cities have significant spatial correlations, presenting the spatial agglomeration characteristics of "high-high" and "low-low".

## Introduction

In recent years, with the rapid development of industrialization and urbanization, the living environment of mankind has been dramatically improved, the material foundations have become more abundant, and billions of people have shaken off poverty. However, these economic prosperities lie in the sacrifice of the environment (Zhong et al., 2021 [1]). In the past, the development patterns of high investment, high consumption, and high pollution have led to many severe ecological problems, such as the depletion of natural resources, air pollution, and the destruction of biodiversity (Fan et al., 2023 [2]; Ghafarpasand et al., 2021 [3]; Kassouri, 2021 [4]). In addition, unbalanced regional development has caused many social problems,

(22&ZD156 to SX) and Jiaxing University Research Project (70524002 to LL).

**Competing interests:** The authors declare that no competing interests exist.

such as regional development gaps, wealth gaps, and uneven development opportunities (Sulemana et al., 2019 [5]). In this context, the concept of inclusive green growth was first proposed at the Rio+20 Summit in 2012. Subsequently, the World Bank formally proposed this concept, emphasizing that industrial economic growth must consider inclusiveness and greenness. This concept offers a new development direction for many developing countries and has received widespread attention in the academic community. A rich body of literature has emerged focusing on the concept studies of inclusive green growth, the measuring studies of the inclusive green growth performance level (IGGPL), and the studies on the factors affecting IGGPL.

When discussing the concept of inclusive green growth, we may encounter two schools of thought. One emphasizes the importance of greenness for inclusive green growth from the perspective of development economics. This viewpoint holds that, in terms of long-term economic development, if economic growth lacks greenness, it cannot be sustainable (Wellmann et al., 2020 [6]; Spratt, 2013 [7]). In other words, only green economic growth can ensure that our limited natural assets continue to provide the necessary resources and environmental services for human economic activities, thereby promoting sustainable economic growth and meeting the development needs of the world's poor population. Therefore, to promote sustained economic growth and ensure the long-term welfare of future generations (Quaas and Smulders, 2018 [8]), there needs to be a decoupling between economic growth and the consumption of natural resources, meaning economic growth should be achieved with minimal consumption of natural resources. In practice, countries need to propose reasonable environmental policies based on their actual conditions (Meyer et al., 2012 [9]) and encourage green technological innovation, develop sustainable energy sources, and shape sustainable cities to enhance resource efficiency, reduce pollutant emissions, and protect biodiversity (Jacobs, 2012 [10]), ensuring sustainable development is realized. The other school of thought emphasizes the importance of inclusiveness for inclusive green growth in terms of welfare economics. This viewpoint holds that the starting point and the goal of economic growth are to promote the comprehensive development of people. Economic growth should consider not only future human well-being but also current human welfare (Kaur and Garg, 2019 [11]), which requires us to employ sustained economic growth to address many inequality issues in social development, such as widening wealth gaps, unequal development opportunities, development achievements that cannot be equally distributed, and weak social cohesion (Albagoury, 2016 [12]; Spanish, 2020 [13]; Khoday and Perch, 2012 [14]). Therefore, some scholars believe that inclusive green growth not only means sustained economic growth but more importantly it is about creating new development opportunities through economic growth (Ali and Zhuang, 2007 [15]; Sugden, 2012 [16]). These new development opportunities enable more people to participate in economic activities, meet the development needs of the poor, and allow more people to share the fruits of economic development (Albagoury, 2016 [12]; Kaur and Garg, 2019 [11]). Moreover, some scholars pointed out that emphasizing the inclusiveness of inclusive green growth can protect the basic welfare of the poor (Spanish, 2021 [13]), reduce the wealth gap, enhance social cohesion (Quaas and Smulders, 2018 [8]; Hu and Wang, 2019 [17]), and thus promote long-term sustainable economic growth. The academic community has not reached a consensus on the concept of inclusive green growth. Inclusive green growth needs to consider the balance between long-term and short-term human well-being, as well as the balance between economic growth and environmental sustainability, and it also needs to consider the balance between economic growth and social equity.

Regarding the research on measuring IGGPL, the academic community has mainly developed two types of literature. One type of literature focuses on exploring IGGPL from the perspective of methodological improvement. For instance, Chen et al. (2020) [18] combined the super-efficiency slack-based measure model with the meta-frontier Malmquist-Luenberger

index approach to assess the IGGPLs of 108 cities in the China's Yangtze River economic belt. Simultaneously, Sun et al. (2020) [19] used the comprehensive directional distance function and the model based on slack variables to analyze the IGGPLs of 285 Chinese cities from 2003 to 2015. Ren et al. (2022) [20] integrated the slack-based measure model of directional distance function with the global Malmquist-Luenberger index to calculate the IGGPL of 282 Chinese cities from 2004 to 2019. The other type of literature focuses on measuring the IGGPL from the perspective of constructing evaluation indicator systems. This body of literature can be mainly divided into two branches: one branch emphasizes the construction of the evaluation indicator system for comparing the IGGPLs between countries. Representative works include Kumar (2015) [21], who proposed an inclusive green growth indicator system for the Association of Southeast Asian Nations from the perspectives of livability, intelligence, and low carbon. The Asian Development Bank in 2018 constructed an evaluation indicator system with 28 indicators based on economic growth, social equality, and environmental sustainability, and assessed the IGGPLs of 45 developing countries and regions in Asia. Herrero et al. (2020) [22] established an inclusive green energy progress indicator system including inclusiveness, greenness, and energy efficiency, and used the data from 2004–2014 for 157 countries to evaluate the progress of inclusive green energy. Halkos et al. (2021) [23] created a comprehensive index system that captures the industrial socio-economic inclusiveness and greenness of various world economies and conducted a comparative analysis of 83 economies in 2016. This branch of literature allows for international comparison of indicators and enables a country to find the differences in its IGGPL compared to other countries IGGPLs. The other branch of literature mainly constructs an indicator system capable of evaluating IGGPL within a particular country or region. For example, Albagoury (2016) [12] designed an indicator system from the perspectives of economic growth, productive employment, economic infrastructure, income equity, and social inclusiveness, and measured IGGPL in Ethiopia. Kumar (2017) [24] showed Japan's inclusive wealth index by constructing an indicator system that includes human capital, produced capital, and natural capital stocks. Zhou (2022) [25] established an indicator system from the following four dimensions: economic development, equal opportunities, green production and consumption, and ecological environment protection, and calculated the IGGPLs for China's 31 provinces from 2010–2019. Zhang et al. (2021) [26] constructed an indicator system of the Chinese IGGPL from 12 dimensions including per capita material footprint, air pollution, renewable energy supply, Palma ratio, and basic service access.

Existing studies on the impact factors of IGGPL mainly focus on the economy and policy factors. Firstly, IGGPL is greatly influenced by various economic activities, including positive, negative, and uncertain impacts. In terms of positive impacts, some scholars believe that the upgrading of industrial structures can lead to more efficient resource utilization and less environmental pollution (Auty, 1997 [27]), which is conducive to improving the quality and efficiency of economic growth, creating more job opportunities, and increasing workers' income, thereby promoting IGGPL. There is also literature confirming that technological innovation (Fankhauser and Bowen, 2013 [28]), digital inclusive finance (Dara and Rao 2018 [29]), and digital transformation (Liu et al., 2018 [30]) can effectively enhance IGGPL. Regarding negative impacts, some studies suggested that excessive house price inflation not only increases the housing costs for low-income people, compressing their consumption expenditure in other areas (Charles et al., 2018 [31]) but may also exacerbate social inequality (Adelino et al., 2015 [32]), which means excessive house price inflation is not conducive to enhancing IGGPL. In terms of uncertain impacts, some literature pointed out that industrial agglomerations can reduce production costs for enterprises, decrease energy consumption and emissions during production (Fujita and Thisse, 2003 [33]), provide economies of scale (Andersson and Lööf, 2011 [34]), and thus promote IGGPL. However, industrial agglomerations might also lead to

regional environmental degradation, hinder balanced regional developments, and damage regional IGGPLs. Concerning policy impacts, reasonable institution design often promotes IGGPL. For instance, the reasonable minimum wage system can guarantee the basic income of the low-income labor group, meet the basic living needs of the low-income group (Gan et al., 2016 [35]), and help to narrow wealth gaps. Formal environmental regulation systems may not directly affect IGGPL in the short term, but their impacts on IGGPL will gradually emerge over time (Telle and Larsson, 2007 [36]; Jorgenson and Wolcoxen, 1990 [37]). Additionally, land fiscal systems affect the distribution and utilization of land resources. A reasonable land fiscal system can increase public welfare, such as parks and green spaces, but over-reliance on land transfer incomes as fiscal sources is not a sustainable development pattern, which is detrimental to IGGPL.

The existing literature has conducted a significant amount of work on the concept of inclusive green growth, the statistical measurement of IGGPL, and the factors affecting IGGPL. This research plays an important role in scientifically achieving the essence of inclusive green growth. According to the existing studies on the inclusive green growth concept, this paper believes that inclusive green growth is a balanced growth pattern that aims to enhance the welfare of both current and future generations and achieves stable economic growth under the conditions of minimizing the consumption of natural assets and maximizing social equity, ultimately promoting sustainable development. In addition, due to differences in national backgrounds, population status, resource endowments, and institution designs among different countries, the strategies for promoting inclusive green growth vary from country to country. As the largest developing country, China is undergoing the crucial stage of economic structural adjustment, and many deep-seated issues in the traditional economic developments have gradually emerged, leading to problems such as unbalanced and inadequate economic development. In terms of economic growth, issues such as weakening domestic demand, declining investment, aging populations, and decreasing birth rates have increased the downward pressures on economic growth (Huang et al., 2012 [38]). In terms of ecological environments, China is currently facing some environmental challenges such as air pollution, water contamination, land degradation, and localized ecological destruction (Caglar, et al, 2024 [39]). Regarding social inclusion, although China's impoverished population had been completely lifted out of poverty by 2020, there are still significant disparities in income distribution, educational opportunities, health care, and employment among different groups such as between urban and rural areas, across different regions, and between registered urban residents and the floating population (Song et al., 2024 [40]; Dong et al., 2022 [41]). These issues of inadequacy and imbalance in economic development undoubtedly demand an acceleration in the green transformation of the economic structure towards high-quality development. Inclusive green growth thus has become a necessary option for China's future economic development. However, research on China's IGGPL is mainly focused on provinces, economic zones, or city clusters, with fewer studies exploring the evolution characteristics of IGGPL at prefecture-level cities. There is a lack of a relatively objective evaluation index system for assessing the IGGPLs of cities in China, and fewer studies have conducted in-depth analyses of the spatial correlations and spatial agglomeration characteristics of China's city-level IGGPLs. Moreover, scholars have rarely offered the underlying reasons behind the evolutionary characteristics of China's IGGPL from a quantitative perspective.

Given these, this article constructs an evaluation index system for IGGPL in Chinese cities and utilizes panel data from 271 Chinese cities from 2006 to 2020 to explore the temporal and spatial evolutions, distribution dynamics, regional disparities, and spatial correlations of IGGPLs at the city level. The marginal contributions of this article are as follows. First, according to the unique economic-social-environmental characteristics of Chinese cities, we design

an evaluation index system for inclusive green growth at the prefecture-level city that considers China's conditions, enriching the theory of measuring IGGPL and being in line with China's current goals of achieving peak carbon emissions and carbon neutrality. Second, we provide comprehensive analyses of the temporal and spatial evolutions, distribution dynamics, and regional disparities of the IGGPLs at the city level in China by using a variety of statistical methods. Third, we investigate the transition probabilities of IGGPLs, uncovering the intrinsic reasons behind the formation of the evolutionary characteristics of IGGPLs in Chinese cities.

## Indicator system

### The characteristics of IGGPL

Based on the above descriptions of IGGPL, we argue that the inclusive green growth is essentially a balanced growth pattern that takes the enhancement of the well-being of present and future generations as its value orientation and achieves stable economic growth under the conditions of minimizing the consumption of natural assets and maximizing social equity, ultimately promoting sustainable development. It emphasizes the coordination of economic growth, social inclusion, and environmental greening. Given these, we summarize the three keys of inclusive green growth as follows.

One key is the economic growth (Ali and Zhuang, 2007 [15]; Sugden, 2012 [16]). Economic growth is the starting point for inclusive green growth. The sustainable development of human beings cannot be achieved without solid material bases. Only sustainable economic growth can create more social wealth and employment opportunities, thus providing a solid material guarantee for the inclusive development of human society. The second key is the social inclusion (Spanish, 2021 [13]; Quaas and Smulders, 2018 [8]; Hu and Wang, 2019 [17]). Through the promotion of fair opportunities for members of society in such areas as employment, health care, education, social security, and public resources, the growing problem of social inequality will be resolved so that people can equally participate in the development process and enjoy the fruits of development, thereby ultimately realizing the goal of sustainable development. The third key is the environmental greening (Wellmann et al., 2020 [6]; Spratt, 2013 [7]). Environmental greening is a foundation of inclusive green growth. The realization of sustainable development needs to fully consider the coordinated development of human beings and nature so that limited natural assets can continuously serve human development. By enlarging green ecological space, relaxing environmental pressure, and strengthening the management of pollution emissions, the negative impacts of human production activities on the ecological environment will be effectively mitigated, the over-exploitation of natural resources will be avoided, and natural assets will be able to provide human beings with the resources and environment on which they depend in the long term, thereby ultimately realizing sustainable development.

### Indicator system construction of IGGPL

Based on the concept of IGGPL, combining existing literature, and considering the correlations and measurability of indicators, we construct an indicator system of IGGPLs in Chinese prefecture-level cities (see Table 1). This indicator system construction follows strict procedures, and the resulting indicator system can also describe sub-dimensions well. The effectiveness and robustness of the indicator system are verified by the entropy weighting method and the Criteria Importance Through Intercrieria Correlation (CRITIC) weighting method. In addition, all indicators in Table 1 pass the Pearson correlation coefficient test, with weak correlations between indicators and no redundant indicators.

**Table 1. Indicator system of IGGPLs in Chinese prefecture-level cities.**

| Dimension | Sub-dimension | Indicator name | Unit | Nature | Weight |
|---|---|---|---|---|---|
| Economic growth performance | Level of economic growth [23, 25] | GDP per capita | Yuan/person | + | 0.0334 |
| | | Government revenue | Ten thousand yuan | + | 0.1266 |
| | Quality of economic growth [10, 18] | Industrial structure superiority | % | + | 0.0095 |
| | | Percentage of employees in the productive service industry | % | + | 0.0198 |
| Social inclusion performance | Equity in basic security [2] | Water penetration rate | % | + | 0.0030 |
| | | Gas penetration rate | % | + | 0.0032 |
| | | Per capita electricity consumption | kWh/person | + | 0.0633 |
| | Equity in public resources [2] | Number of books in public libraries per 100 people | Volumes, pieces/100 persons | + | 0.1133 |
| | | Number of buses, trams, and cabs per 10,000 people | Vehicles/ million people | + | 0.0741 |
| | | Urban road area per capita | Square meter/person | + | 0.0171 |
| | Income and consumption equity [20] | Real per capita disposable income of urban residents | Yuan | + | 0.0240 |
| | | Real retail sales of consumer goods per capita | Yuan/person | + | 0.0486 |
| | | Unemployment rate | % | - | 0.0016 |
| | Equity in social security [2] | Pension insurance participation rate | % | + | 0.0666 |
| | | Medical insurance participation rate | % | + | 0.0642 |
| | | Unemployment insurance participation rate | % | + | 0.0912 |
| | Healthcare equity [18, 20] | Number of hospitals per 10,000 people | Per 10,000 people | + | 0.0549 |
| | | Number of hospital beds per 10,000 people | Sheets/ million people | + | 0.0213 |
| | | Number of doctors per 10,000 people | Persons/million | + | 0.0289 |
| | Education equity [20] | Compulsory education teacher-student ratio (primary and middle school) | Persons/million | + | 0.0327 |
| | | Real per capita public expenditure on education | Yuan/person | + | 0.0428 |
| Environmental greening performance | Green living space [2, 23] | Greening coverage rate of built-up areas | | + | 0.0047 |
| | | Park green space per capita | Square meter | + | 0.0158 |
| | | Annual average concentration of PM2.5 | ug/m3 | - | 0.0098 |
| | Pollution discharge emission control [20] | Industrial wastewater emission per unit of GDP | Ton/yuan | - | 0.0011 |
| | | Industrial sulfur dioxide emissions per unit of GDP | Ton/ten thousand yuan | - | 0.0017 |
| | | Industrial fume and dust emissions per unit of GDP | Ton/ten thousand yuan | - | 0.0015 |
| | | Carbon dioxide emissions per unit of GDP | Ton/yuan | - | 0.0028 |
| | Environmental pollution control and remediation [19, 20] | Centralized treatment rate of the sewage treatment plant | % | + | 0.0072 |
| | | Harmless treatment rate of domestic garbage | % | + | 0.0079 |
| | | The comprehensive utilization rate of general industrial fixed waste | % | + | 0.0075 |

Note: Superiority of industrial structure = Value-added of the primary industry as a share of GDP1+Value-added of the secondary industry as a share of GDP2+Value-added of the tertiary industry as a share of GDP3.

## Methods

### Samples and data sources

We select the panel data of 271 cities in the Chinese mainland from 2006 to 2020. Some statistical indicators before 2006 are unsound and seriously missing. During 2011–2019, China made

some adjustments to the division of urban administrative regions. Many cities are abolished, and there are also some new cities born. Therefore, these cities are not in our study. The data is from the China Urban Statistical Yearbook, the China Regional Statistical Yearbook, the China Urban and Rural Statistical Yearbook, the China Energy Statistical Yearbook, and the EPS database. The PM2.5 data is from the Center for Socioeconomic Data and Applications at Columbia University in the United States, and the missing data is solved by interpolation.

## VHSD method

Determining the weights of indicators reasonably is key to statistical evaluation research. The common methods include objective weighting methods (e.g., the entropy method, TOPSIS method, CRITIC method, and principal component analysis method), subjective weighting methods (e.g., the Delphi method, AHP method, Least Squares method, and Equal Weights method), and the combined subjective and objective approach. Objective weighting methods are usually not applicable because they focus more on individual subjective consciousnesses. Objective weighting methods avoid this drawback, but the panel data weights calculated by them cannot make dynamic comparisons across periods. Given these, we adapt the VHSD method, allowing us to conduct dynamic comparisons across periods objectively.

Suppose $s_1, s_2, \ldots, s_n$ are $n$ evaluated objects, $x_1, x_2, \ldots, x_m$ are $m$ evaluation indicators, the vector of indicator weights is $W = (\omega_1, \omega_2, \ldots, \omega_m)'$, and $t_1, t_2, \ldots, t_T$ are times. Then, the panel data studied in this paper can be represented as $\{x_{ij}(t_k)\}$, $i = 1, 2, \ldots, n, j = 1, 2 \ldots, m$, and $k = 1, 2 \ldots, T$, where $x_{ij}(t_k)$ is the raw value of the $x_j$-th evaluated object. The original data is standardized by

$$x^*_{ij} = \begin{cases} (x_{ij}(t_k) - \min_j)/(\max_j - \min_j) & x_{ij}(t_k) > 0 \\ (\max_j - x_{ij}(t_k))/(\max_j - \min_j) & x_{ij}(t_k) \leq 0 \end{cases} \tag{1}$$

where $\max_j$ ($\min_j$) means the maximum (minimum) of the indicator $x_j$. Constructing the evaluation function

$$IGGPL_i(t_k) = \sum_{j=1}^{m} \omega_j x^*_{ij}(t_k), i = 1, 2 \ldots, n, j = 1, 2 \ldots, m, k = 1, 2 \ldots, T. \tag{2}$$

The general principle of the VHSD method for determining the weights $\omega_j, j = 1, 2 \ldots, m$ is maximizing the differences between objects. That is, the sum of squares of deviations of $IGGPL_i(t_k)$ ($\sigma^2$ for short) is maximized, i.e.,

$$\sigma^2 = \sum_{k=1}^{T} \sum_{i=1}^{n} (IGGPL_i(t_k) - \overline{IGGPL}). \tag{3}$$

The original data is standardized. Thus

$$\overline{IGGPL} = \frac{1}{T} \sum_{k=1}^{T} \left[ \frac{1}{n} \sum_{i=1}^{n} \sum_{j=1}^{m} \omega_j x^*_{ij}(t_k) \right] = 0. \tag{4}$$

Then

$$\sigma^2 = \sum_{k=1}^{T} \sum_{i=1}^{n} (IGGPL_i(t_k))^2 = \sum_{k=1}^{T} [W^T H_k W] = W^T \sum_{k=1}^{T} H_k W = W^T H W. \tag{5}$$

where $H = \sum_{k=1}^{T} H_k$, $H$ is a symmetric matrix of order $m \times m$, $H_k = X_k^T X_k$, $k = 1, 2, \ldots, T$, and

$$X_k = \begin{bmatrix} x_{11}^*(t_k) & \cdots & x_{1m}^*(t_k) \\ \cdots & \cdots & \cdots \\ x_{n1}^*(t_k) & \cdots & x_{nm}^*(t_k) \end{bmatrix}, \; k = 1, 2 \ldots, T. \tag{6}$$

To avoid $\sigma^2$ being too large, we set $||W|| = \omega_1^2 + \omega_2^2 + \cdots + \omega_m^2 = 1$. Under this set, we can prove that when $W$ is equal to the eigenvector corresponding to the largest eigenvalue of the symmetric matrix $H$, $\sigma^2$ is largest. In addition, we set $W > 0$ to ensure that weights are positive. Then, the weighting coefficient vector can be computed by solving the linear programming:

$$\max \; W^T H W \quad \text{s.t.} \quad ||W|| = 1, W > 0. \tag{7}$$

## Kernel density method

Kernel density estimation is a method for analyzing spatial imbalances, which can used to graphically describe the dynamic evolution of IGGPL distribution. Denoted by $f(y)$ the density function of IGGPL ($y$ for short), we construct the equation:

$$f(y) = \frac{1}{N\rho} \sum_{i=1}^{N} K\left(\frac{Y_i - y}{\rho}\right) \tag{8}$$

with

$$K(y) = \frac{1}{2\pi} \exp(-y^2/2), \tag{9}$$

where $N$ is the number of prefecture-level cities, and $K(\cdot)$ is the kernel function with the bandwidth $\rho$. In this paper, we adopt the common Gaussian kernel function. $Y_i$ is independent identical distribution observations.

## Dagum Gini coefficient method

The classical Gini coefficient method is commonly used to measure regional imbalances. However, it works well only when the data is homoscedastic and normally distributed. The Dagum Gini coefficient method (Dagum, 1997) [6] relaxes these assumptions and has stronger applicability. Dagum Gini coefficient method divides the overall Gini coefficient of IGGPL ($G$ for short) into three parts, including intra-regional difference ($G_\omega$), inter-regional net difference ($G_{nb}$), and intensity of trans-difference ($G_t$). Then $G = G_\omega + G_{nb} + G_t$, where

$$G = \left(\sum_{j=1}^{k}\sum_{h=1}^{k}\sum_{i=1}^{n_j}\sum_{r=1}^{n_h}|Y_{ji} - Y_{hr}|\right)/(2n^2\overline{Y}), \tag{10}$$

$$G_\omega = \sum_{j=1}^{k} G_{jj}p_j s_j, \tag{11}$$

$$G_{nb} = \sum_{j=2}^{k}\sum_{h=1}^{j-1} G_{jh}(p_j s_h + p_h s_j)D_{jh}, \tag{12}$$

$$G_{t} = \sum_{j=2}^{k}\sum_{h=1}^{j-1} G_{jh}(p_j s_h + p_h s_j)(1 - D_{jh}), \tag{13}$$

$$G_{jj} = (\sum_{i=1}^{n_j}\sum_{r=1}^{n_h}|Y_{ji} - Y_{hr}|)/(2n_j^2\overline{Y}_j), \tag{14}$$

$$G_{jh} = (\sum_{i=1}^{n_j}\sum_{r=1}^{n_h}|Y_{ji} - Y_{hr}|)/(2n_j n_h(\overline{Y}_j + \overline{Y}_h)), \tag{15}$$

In Eqs (10)–(15), the number of regions $k$ = 4, the number of cities $n$ = 271, $h$ and $j$ denote the region code (1, 2, 3, 4), $n_j(n_h)$ is the number of cities in the $j(h)$ region, and $Y_{ji}$ $(Y_{hr})$ is the IGGPL in the $j(h)$ region. In addition, $\overline{Y}$ represents the average national IGGPL, $\overline{Y}_j(\overline{Y}_h)$ and represents the average IGGPL in region $j(h)$. $G_{jj}$ is the Gini coefficient in region $j$, and $G_{jh}$ is the Gini coefficient between regions $j$ and $h$. We denote by $p_j = n_j/n$, $p_h = n_h/n$, $s_j = (n_j\overline{Y}_j)/(n\overline{Y})$, and $s_h = (n_h\overline{Y}_h)/(n\overline{Y})$. $D_{jh}$ represents the relative impact of IGGPLs between the $j$-th and $h$-th regions, i.e., $D_{jh} = (d_{jh} - p_{jh})/(d_{jh} + p_{jh})$. When $Y_{ji} - Y_{hr} > 0$, $d_{jh}$ represents the expectation of the sum of all sample values between regions $j$ and $h$, i.e., $d_{jh} = \int_0^{\infty} dF_j(y) \int_0^{y}(y - x)dF_h(x)$. When $Y_{hr} - Y_{ji} > 0$, $p_{jh}$ represents the expectation of the sum of all sample values between regions $j$ and $h$, i.e., $p_{jh} = \int_0^{\infty} dF_h(y) \int_0^{y}(y - x)dF_j(x)$. $F_j(x)$ and $F_h(x)$ are the cumulative density distribution functions of IGGPLs in regions $j$ and $h$, respectively.

## Moran index method

Total Moran Index can reveal the total difference among cities within a country, which is expressed as

$$I = \frac{\sum_{i=1}^{n}\sum_{j=1}^{n} W_{ij}(Y_i - \overline{Y})(Y_j - \overline{Y})}{S^2 \sum_{i=1}^{n}\sum_{j=1}^{n} W_{ij}}, \tag{16}$$

where $S^2 = (1/n)\sum_{i=1}^{n}(Y_i - \overline{Y})^2$, $\overline{Y} = (1/n)\sum_{i=1}^{n}Y_j$, $Y_j$ is the IGGPL in the $i$-th region, $n$ is the number of regions, and $W_{ij}$ is the spatial weight matrix. We adopt the 0–1 spatial weight matrix. $I \in [-1, 1]$, where $I \in (0, 1]$ ($I \in [-1, 0)$) means positive (negative) spatial correlation. The larger the $|I|$ is, the stronger the spatial correlation will be. If $I = 0$, the spatial correlation is zero. In addition, the formula for local Moran index is

$$I_i = [(Y_i - \overline{Y})/S^2]\sum_{j} W_{ij}(Y_j - \overline{Y}), \tag{17}$$

which can reveal the difference among cities within a region.

## Markov chain method

A Markov chain, denoted by $\{Y(t), t = 1, 2, \ldots, T\}$, is a set of the discrete random variables with Markov properties. By analyzing the state transfer probability matrix of these random variables, Markov analyses can reveal the state transfer features of IGGPLs of Chinese prefecture-level cities. The state transfer probability distribution of the IGGPL in year $t$ can be represented by the vector $P^t$, where $P^t = (P_1^t, P_2^t, \ldots, P_k^t)'$, and the state transfer probability distributions of IGGPLs in different years can be represented by a matrix $M$ of order $k \times k$. The element in $M$ is

$$P_{i,j}^{t,t+d} = P(Y_{t+d} = j | Y_{t+d} = i) = \frac{\sum_{t=2006+d}^{2020} n_{i,j}^{t,t+d}}{\sum_{t=2006}^{2020-d} n_i^t}, \tag{18}$$

where $P_{i,j}^{t,t+d}$ is the probability of the IGGPL of a region transfers from state $i$ in year $t$ to state $j$ in year $t + d$, $n_{i,j}^{t,t+d}$ is the corresponding number of cities, and $n_i^t$ is the number of cities whose IGGPL is in state $i$ in year $t$. The spatial Markov chain develops the traditional Markov chain, which can explore the state transfer probability of the local IGGPL under the consideration of the impacts of neighboring cities' IGGPLs.

## Empirical results and analyses

### Characteristic analyses of IGGPL

Based on the index system constructed above, we employ the VHSD method to measure the IGGPLs of 271 prefectural-level cities in China from 2006 to 2020. The used IGGPL values are 10 times their original value. According to quantiles, we divide them into four intervals, i.e., [0.001, 0.783), [0.784, 0.951), [0.952, 1.191), and [1.192, 5.602), which correspond to the low level (Q1), the medium-low level (Q2), the medium-high level (Q3), and the high level (Q4), respectively. Fig 1 shows the number of cities at each IGGPL level from 2006 to 2020. In 2006, the numbers of cities at Q1, Q2, Q3, and Q4 were 174, 40, 34, and 23, respectively, indicating that the IGGPLs of this period were still low. The number of cities at Q4 became 69 in 2013, which is significantly larger than the number of cities at Q1. Up to 2020, the numbers of cities at Q3 and Q4 have been 116 and 111, respectively, while the number of cities at Q1 has dropped to zero, indicating that the IGGPLs of most cities in China have been fundamentally improved. The explanation for this phenomenon is that in November 2012, the Chinese

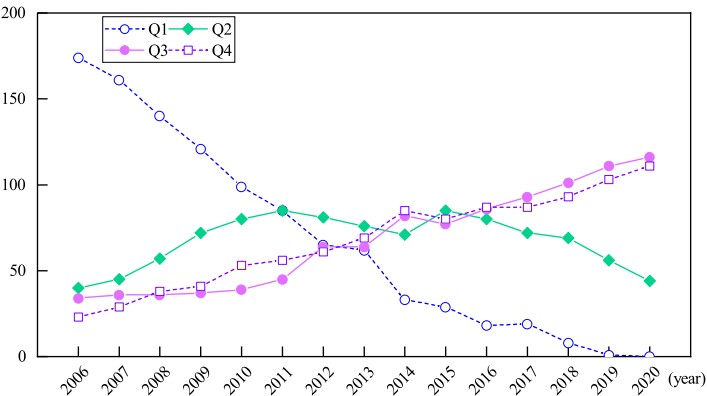

**Fig 1. The number of cities at each IGGPL level from 2006 to 2020.**

government issued the policy "Guiding Opinions on Accelerating Green Development and Building a Beautiful China". Under the guidance of this policy, governments at all levels in China actively promote the optimization and upgrading of industrial structures. In the process of economic development, they not only focus on environmental protection and resource conservation but also attach importance to improving people's livelihoods and reducing poverty, which has positive impacts on the environment and society.

To examine the spatial dynamic evolutions of IGGPLs, according to the economic zones in China, we divide the 271 cities into four regions, including the east, central, west, and northeast regions. Fig 2 shows the national and regional IGGPLs throughout 2006–2020. In terms of the national average, China's IGGPL showed a gradual upward trend throughout 2006–2020, roughly experiencing two critical development stages. The first stage was from 2006 to 2014, during which the IGGPL of each city proliferated, with an average growth rate of 4.37%. The second stage is from 2015 to 2020, a steady growth stage with an average growth rate of 2.93%. Overall, the average annual growth rate of Chinese IGGPL was 3.57% over this period, indicating a good development trend at the macro level but a slightly weaker development level at the later stage. It can be found that (i) the IGGPLs of the eastern region were the largest, from 1.0054 in 2006 to 1.6223 in 2020, with an average annual growth rate of 3.48%, playing an important role in driving the national IGGPL; (ii) the IGGPL of central and western regions at the beginning of period were almost same, 0.6784 and 0.7208, respectively. However, in terms of growth potential, the average annual growth rate of the IGGPL of the western region, 3.78%, was slightly higher than that of the central region, 3.72%, which suggests that the western region has more significant potential; (iii) compared with the central and western regions, Northeast region, though had the relatively high initial IGGPL (0.8328), has unfortunately experienced a phase of rapid growth and then gradually turned into a low-speed growth stage, of which the 2006–2012 period was a high-speed growth stage with an average growth rate of 4.84%, and the 2013–2020 period turned into a low-speed growth stage with an average growth rate of only 0.90%, with a marked lack of growth potential in the later period.

Fig 3 shows the three sub-dimensions of IGGPL, i.e., the environment, social inclusion, and economic development. Firstly, for national averages for each dimension (see Fig 3(a)), the indices of the three dimensions show a trend of steady growth. The environmental dimension has the highest average index, followed by the social dimension, and finally the economic

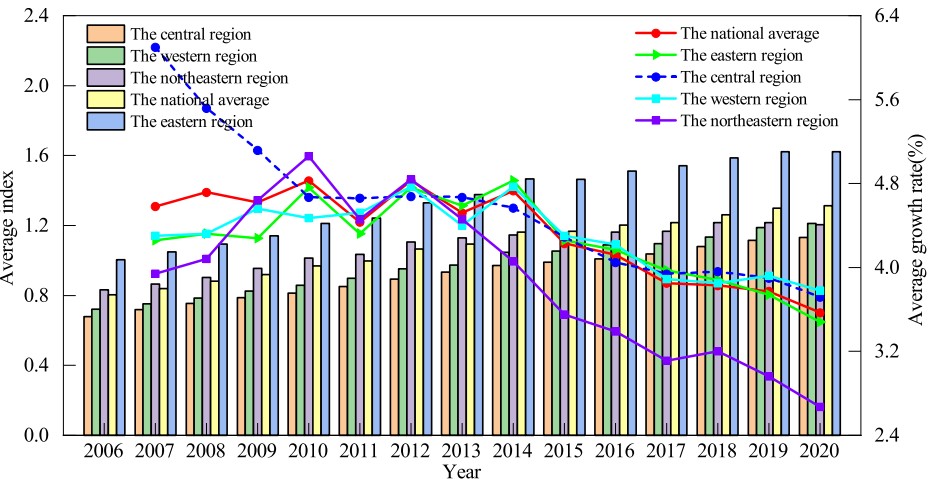

**Fig 2. The national and regional IGGPLs over the period of 2006–2020.**

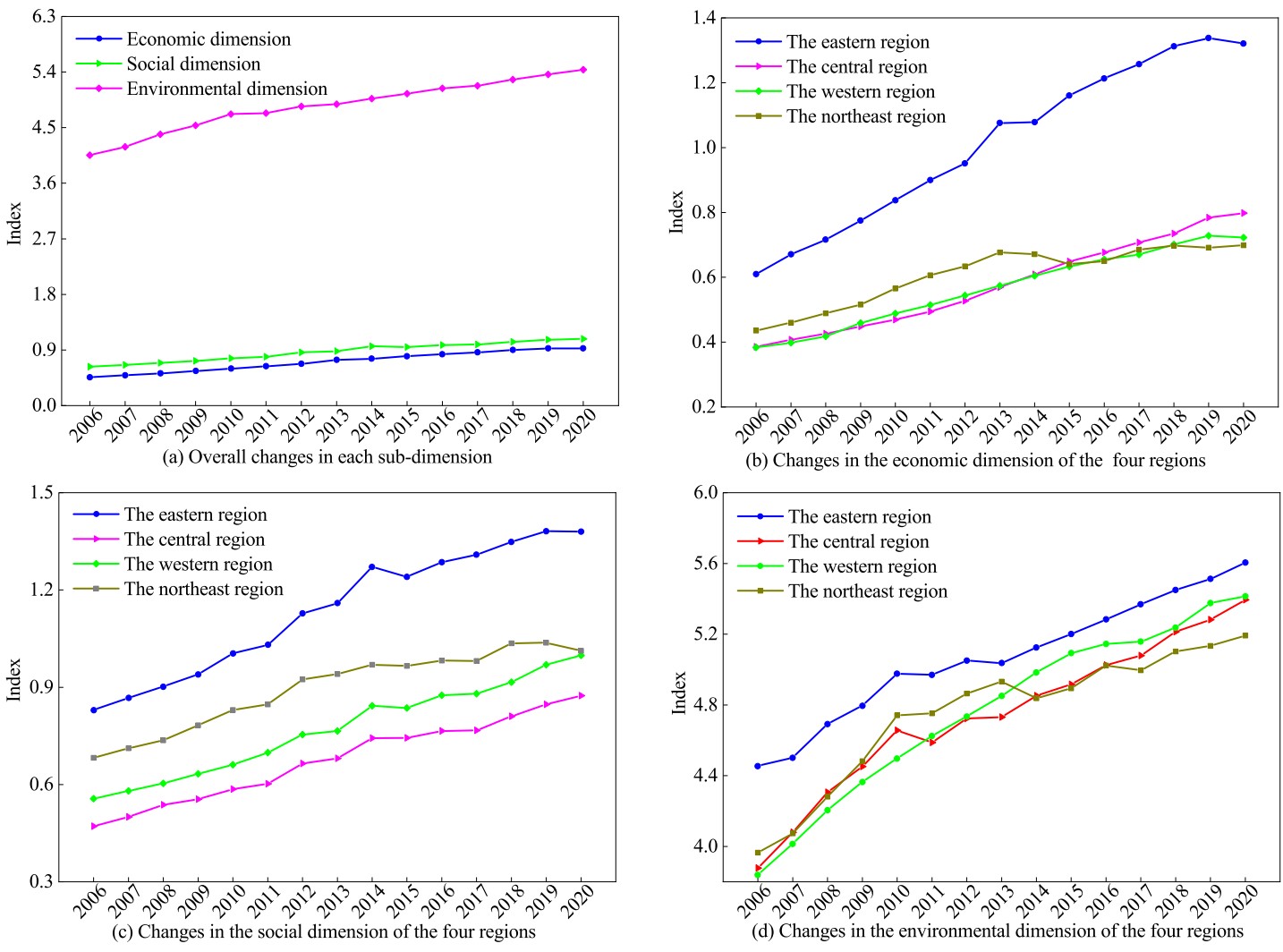

**Fig 3. Different dimension changes of IGGPL in China throughout 2006–2020.**

dimension. As China's economy grows steadily, the Chinese government attaches great importance to protecting the ecological environment and enhancing the well-being of society and people's livelihoods. It has actively and continuously introduced many policies that have greatly contributed to the development of the ecosystem and the social system by proactively transforming the mode of economic growth and unswervingly pursuing development that is innovative, coordinated, green, open, and shared. Secondly, for the economic growth dimension (see Fig 3(b)), the eastern region was far ahead, playing the role of the "leading goose". The indices of the central and western regions in 2006 were roughly equal. However, the growth potential of the central region is significantly higher than that of the western region. The average growth rate of the economic dimension in the central region is 5.34% and is higher than that of the western region (4.62%), which suggests that economic growth is an important advantage of the central region. Although the economic level of the northeastern region in 2006 was higher, it gradually entered a bottleneck after a phase of high growth, and the average annual growth rate of the composite index of the economic dimension was only

3.43%, which was at the bottom of the four major regions, implying that the northeastern region urgently needs to revitalize its economy. Thirdly, for the social inclusion dimension (see Fig 3(c)), the eastern region has a relatively higher level of economic development, and its social governance capacity is stronger and more capable of promoting socially inclusive development. However, it is noteworthy that the northeast region is second only to the eastern region in terms of the social inclusion index, which suggests that the northeast region has been more effective in social inclusion development. In addition, although the social inclusion indexes of the western and central regions are relatively inferior, the average annual growth rates reached 4.50% and 4.26%, respectively, both of which are at a good level, implying stronger potential for future growth. Fourth, for the environment dimension (see Fig 3(d)), in terms of average annual growth potential, the western region has an average annual growth rate of 2.48%, ranking first among the four major regions, and is steadily increasing. The central and northeast regions have average annual growth rates of 2.39% and 1.94%, respectively, ranking 2nd and 3rd. Unfortunately, however, despite its high index in 2006, the eastern region has a lower average annual growth rate of 1.65%, ranking last among the four regions, showing that the lack of momentum in the growth of environmental dimensions in the eastern region.

## Distribution dynamics evolutions

To reveal the dynamic evolution of IGGPLs in the four regions of China, we plot the Kernel density using Matlab2022a software. Fig 4 shows the distribution location, distribution pattern, ductility, and polarization of IGGPLs in the four regions. In terms of distribution location, the center lines of the main peaks of the Kernel density curves in the eastern and central regions are constantly shifted to the right, and the shapes of curves in the two regions are similar, which indicates that their IGGPLs are developing well, are effective, and are showing a steady improvement. The positions of the center lines of the main peaks of the Kernel density curves in the western and northeastern regions have undergone a process of shifting to the left and then to the right, which indicates that their IGGPLs have an evolutionary characteristic of "decreasing first and then increasing". This may be because the two regions are actively pursuing their economic structural transformation in the context of the country's strategy of innovation-driven development and the strategy of "carbon peaking and carbon neutrality", which will lead to a short-lived slight decline in IGGPLs, followed by a sustained improvement.

In terms of distribution morphology, the height of the main peak of the Kernel density curve in the eastern region has a "∩" type of change, and the horizontal width of the curve has slightly contracted, which indicates that the differences in the IGGPLs of the eastern region tend to become smaller. The heights of the main peaks in the central and western regions show the process of "first ∪-type, then ∩-type", which means that the densities of IGGPLs in these two regions have the evolutionary characteristics of "first decreasing, then increasing, and then decreasing". In addition, the horizontal widths of the Kernel density curves of IGGPLs in the two regions vary irregularly, indicating that there is a mixture of convergence and divergence in the degree of intra-regional differences. The height of the main peak of the curve in the northeast region has a fluctuating evolution, but the horizontal width of the curve in the region is gradually narrowing over time, which means that the differences in the urban IGGPLs of the northeast region have a tendency to diminish and the imbalance in development is weakening. In terms of distribution ductility, the Kernel density curves of the four regions are all right trailing, indicating that there exist spatial differences in their IGGPLs. However, in terms of the trailing trend, the ductility of the curves in the eastern, western, and northeastern regions is contracting, which implies that the spatial

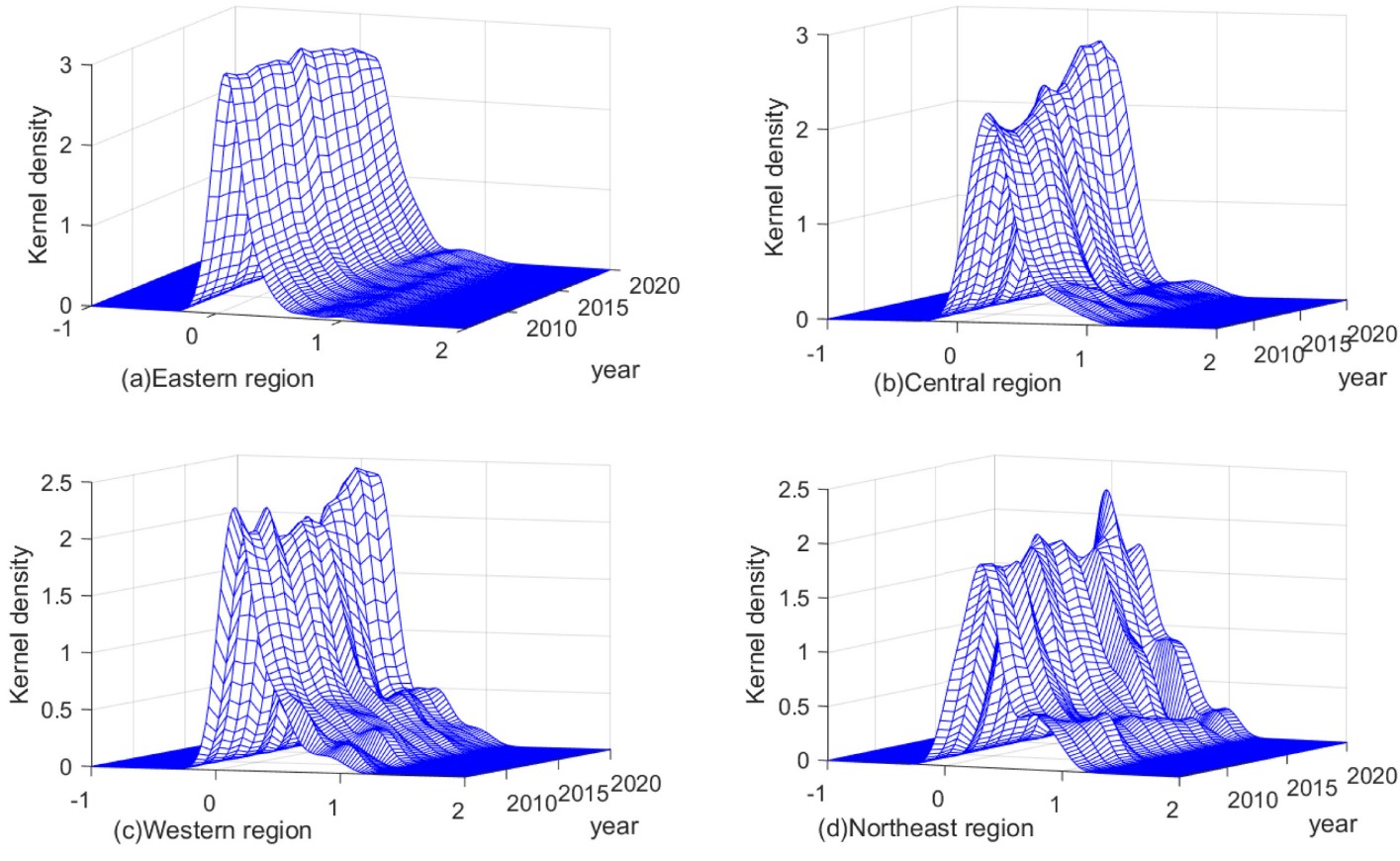

**Fig 4. Kernel density of IGGPLs in the four regions throughout 2006–2020.**

differences in IGGPLs in the three regions are gradually decreasing. The ductility of the curve in the central region has a fluctuating evolutionary trend, which means that the spatial differences in IGGPLs of the central region are changed irregularly and are not expanded significantly on the whole.

In terms of polarization, the eastern and central regions both primarily show one main peak. However, the western and northeastern show the coexistence of main and side peaks, implying a certain degree of divergence in IGGPLs in the two regions. Specifically, the shapes between the main and side peaks in the western region are gradually clear, indicating that polarization is increasing. However, the distance between the main and side peaks is small, and thus there is little differentiation between cities in the region. The shapes between the main and side peaks in the northeast are shifting from very clear to less clear, suggesting that polarization is weakening and that the polarization among cities within the region is disappearing while the development of convergence is increasing. Overall, China's IGGPLs have been improving from 2006 to 2020, but the dynamic evolution of the distributions of IGGPLs in different regions is different. The overall development trends in the eastern and central regions are satisfactory, and the differences between cities within the two regions are relatively small, with no obvious bipolar or multi-polar polarization. In addition, although the western and northeastern have not been able to improve their IGGPLs as fast as the eastern and central regions, there is a certain degree of polarization within the two regions.

### Spatiotemporal difference decomposition

Based on the previous analyses, we know that there are differences in IGGPLs among the four regions of China. However, the sizes of those differences are unknown, the causes of those differences, etc. We analyze these using Dagum Gini coefficients, and their values of them are reported in Fig 5. Fig 5(a) shows the overall differences of IGGPLs. Based on the decline rate of IGGPLs, we divide the period 2006–2020 into two parts: 2006–2012 and 2013–2016. The former has a slower decline rate of IGGPLs, and the latter has a faster rate. The decline rate differences in IGGPLs are clear after 2018. Since the World Bank released Inclusive Green Growth: Pathways to Sustainable Development report in 2012, social inclusiveness and environmental greening have become the consensus for governments to develop their national economies. The Chinese government also attaches great importance to this issue and introduces a number of policies. The implementations of these policies have effectively promoted inclusive green growth in China.

Fig 5(b) and 5(c) illustrate the differences in the IGGPLs of cities within a region, as well as the differences in the IGGPLs of cities between regions. In terms of Fig 5(b), for these four

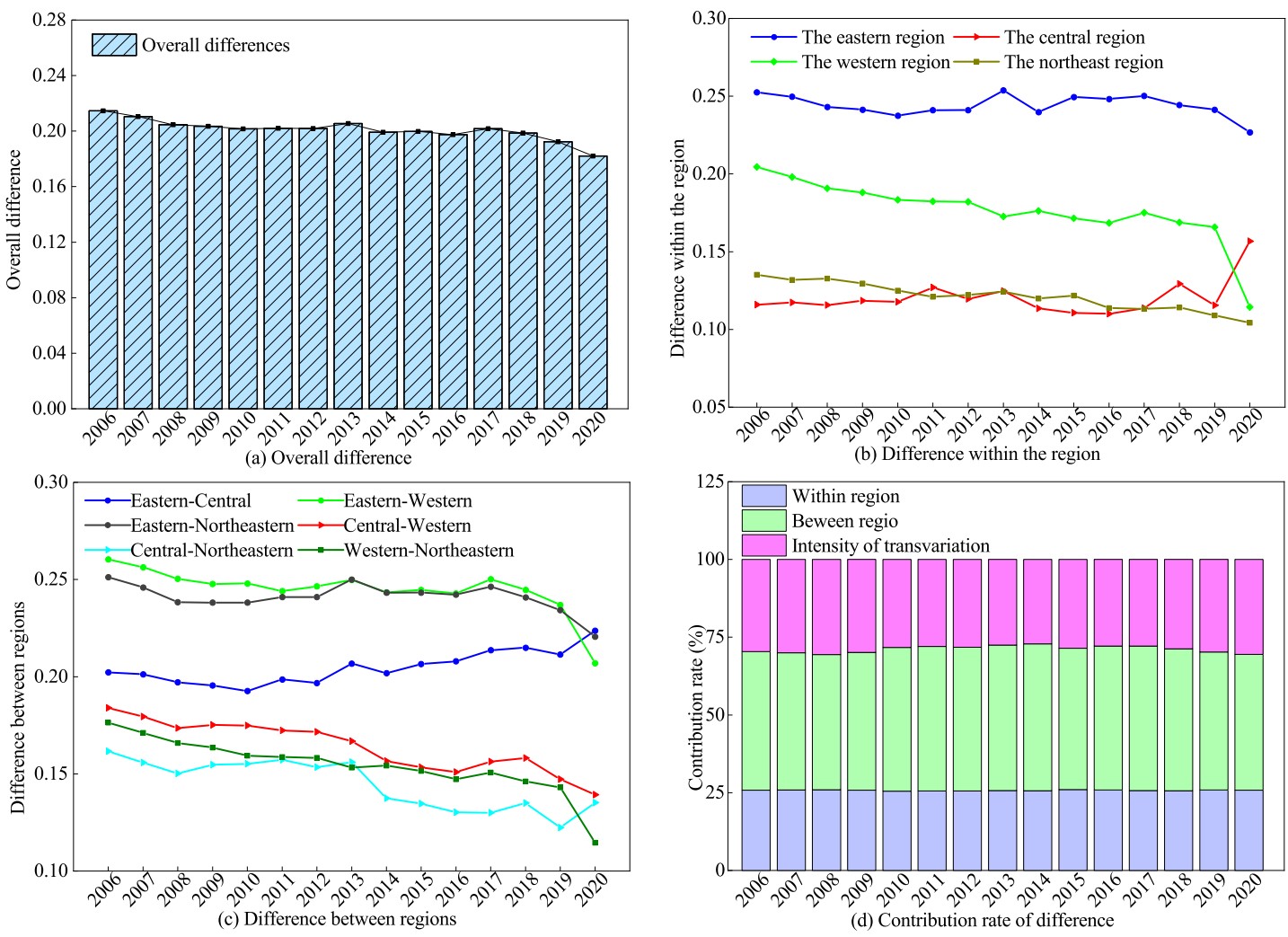

**Fig 5. Regional differences of IGGPLs in China over the period 2006–2020.**

regions, the Gini coefficients in the eastern region are the highest, with a mean value of 0.2440, which indicates that the uneven development of the IGGPLs of the cities in the east region is more serious than that in the other three regions. The east region has many economically developed cities, such as Beijing, Shanghai, Shenzhen, Guangzhou, Suzhou, and Hangzhou. These large cities have a "siphoning effect" on the human, material, and financial resources of neighboring small cities, leading to the unbalanced development of IGGPLs in the cities. The Gini coefficients in the western region are slightly lower than those in the eastern region, with a mean value of 0.1761, which indicates that the IGGPLs of the cities in the western also suffer from some developmental imbalances. The vastness of land, inaccessibility, and the lack of resources in the western region may shackle the communications among cities. In addition, this unbalanced development of IGGPLs in the region is gradually easing after 2019. The mean values of the Gini coefficients in the central and northeastern are smaller (0.1204 and 0.1212, respectively), and the trends of their Gini coefficients are very similar, which suggests that the development of IGGPLs is more balanced in these two regions. The central and northeastern have flatter geographic locations and better transportation conditions, which are conducive to inter-city exchanges. It is noteworthy that the imbalance in the development of IGGPLs in the central region tends to expand after 2018. In terms of Fig 5(c), the differences in the IGGPLs of cities between regions are decreasing. The means of the Gini coefficients in these regions are 0.2448, 0.2409, 0.2047, 0.1639, 0.1542, and 0.1446, respectively. While the Gini coefficients of the "East-Central" region have tended to increase, the Gini coefficients of the other regions have narrowed to varying degrees, suggesting that the imbalance in development between regions is gradually weakening.

Fig 5(d) shows the sources and contribution rates of overall differences. The average contribution rate of inter-regions is the highest, followed by those of hypervariable density and intra-regions, which are 45.42%, 28.84%, and 25.74%, respectively. These imply that the differences in IGGPLs stem mainly from inter-regional differences. The trend in contribution rates shows that the contribution of intra-regional differences has remained stable, from 25.82% in 2006 to 25.81% in 2020, a decrease of only 0.01%. The contribution of inter-regional differences shows a decreasing trend from year to year, from 44.54% in 2006 to 43.69% in 2020, a decrease of 0.85%. Contrary to the former two, the average contribution rate of hypervariable density shows a fluctuating and increasing trend, from 29.64% in 2006 to 30.50% in 2020, an increase of 0.86%.

Based on the analyses above, it is evident that there are differences in IGGPLs among different cities in China, i.e., total differences, and these differences gradually decrease over time. The total differences can be divided into three parts: intra-regional differences, inter-regional net difference differences, and the intensity of trans-differences. Among them, the inter-regional differences are the main reasons for the total differences, and the development gaps between the eastern region and the other three major regions are the key factors causing inter-regional differences. The reasons behind these are that the eastern region has earlier industrial development, faster economic growth, more complete infrastructure and public services, and more investment in social and environmental governance, which can attract a large influx of population, talent, technology, resources, and funds from the central, western, and northeastern regions. These led to faster development in IGGPLs in the eastern region than in the other three regions.

## Spatial correlation and state transition

The above analyses show that IGGPLs in China vary with regions. Can such inter-regional differences cause spatial agglomeration? If so, what are the intrinsic causes? We analyze these

**Table 2. Global Moran's I over the period 2006–2020.**

| Year | 2006 | 2007 | 2008 | 2009 | 2010 | 2011 | 2012 | 2013 |
|---|---|---|---|---|---|---|---|---|
| Moran's | 0.4316 | 0.4504 | 0.4552 | 0.4491 | 0.4594 | 0.4418 | 0.4465 | 0.4386 |
| Z-value | 11.0176 | 11.4132 | 11.4867 | 11.3599 | 11.5988 | 11.1556 | 11.2092 | 11.1897 |
| P-value | 0.0000 | 0.0000 | 0.0000 | 0.0000 | 0.0000 | 0.0000 | 0.0000 | 0.0000 |
| Year | 2014 | 2015 | 2016 | 2017 | 2018 | 2019 | 2020 | |
| Moran's | 0.4429 | 0.4346 | 0.4304 | 0.4144 | 0.3949 | 0.3923 | 0.4145 | |
| Z-value | 11.2022 | 10.9736 | 10.8461 | 10.3914 | 9.8846 | 9.7851 | 10.2911 | |
| P-value | 0.0000 | 0.0000 | 0.0000 | 0.0000 | 0.0000 | 0.0000 | 0.0000 | |

using the Moran index and the Markov chain. Table 2 reports the global Moran's index (Moran's for short) of China's IGGPLs over the period 2006–2020, calculated by using a 0–1 spatial weighting matrix. It can be seen that Moran's is above 0.39, and the P values all pass the significance test. These indicate that there is a significant positive spatial correlation of IGGPLs, i.e., the IGGPLs of a city not only affect those of its surrounding cities but are also affected by the IGGPLs of these surrounding cities.

Fig 6 further reports the local Moran's I, which is used to explore whether there is a local spatial correlation of the IGGPLs. The patterns of "L-L" and "H-H" evidence the strong local spatial correlations. Cities with the "L-L" pattern account for the majority, while those with the "H-H" pattern are the next. In addition, although most cities are in the "L-L" pattern, the number of cities in the "L-L" pattern has decreased significantly over time, while the number of

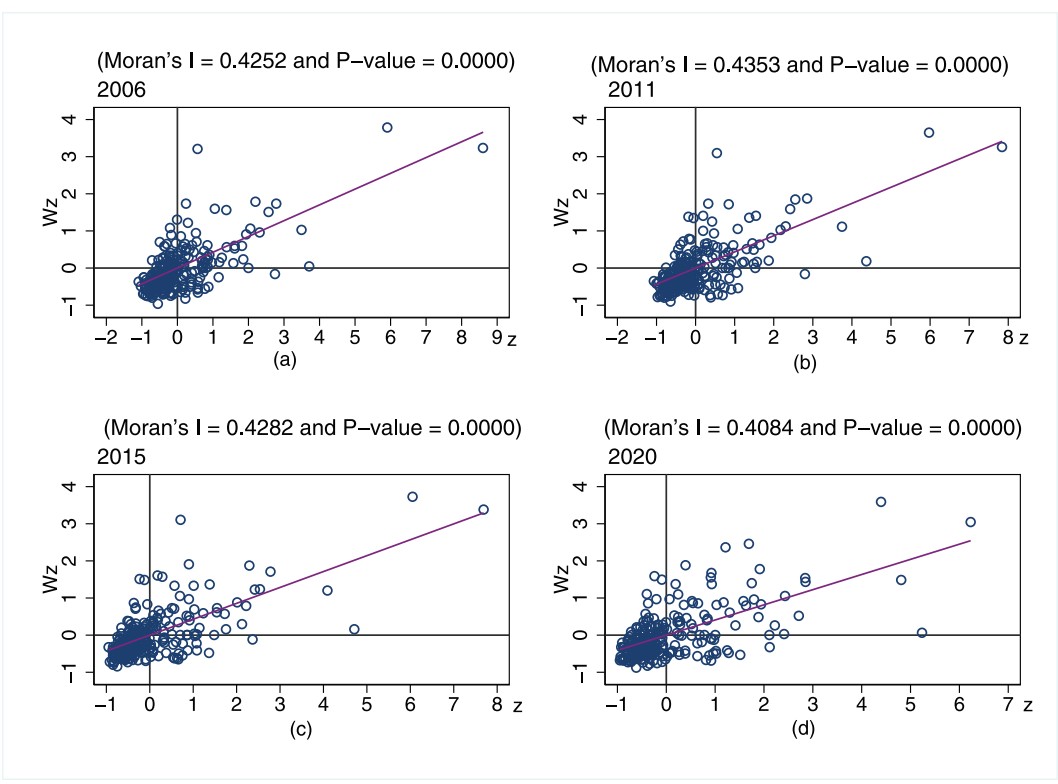

**Fig 6. Local Moran's I over the period 2006–2020.**

**Table 3. Transfer probability matrix of traditional Markov chain.**

| $t \rightarrow t + k$ | Level | $n$ | Q1 | Q2 | Q3 | Q4 |
|---|---|---|---|---|---|---|
| $k = 1$ | Q1 | 1016 | 0.808 | 0.188 | 0.003 | 0.001 |
| | Q2 | 973 | 0.02 | 0.768 | 0.212 | 0.001 |
| | Q3 | 900 | 0 | 0.042 | 0.834 | 0.123 |
| | Q4 | 905 | 0.001 | 0.002 | 0.024 | 0.972 |
| $k = 3$ | Q1 | 1007 | 0.532 | 0.451 | 0.016 | 0.001 |
| | Q2 | 847 | 0.005 | 0.469 | 0.517 | 0.009 |
| | Q3 | 689 | 0 | 0.032 | 0.636 | 0.332 |
| | Q4 | 709 | 0 | 0.004 | 0.025 | 0.97 |
| $k = 5$ | Q1 | 970 | 0.328 | 0.577 | 0.093 | 0.002 |
| | Q2 | 693 | 0.003 | 0.218 | 0.739 | 0.04 |
| | Q3 | 512 | 0 | 0.02 | 0.428 | 0.553 |
| | Q4 | 535 | 0 | 0.006 | 0.024 | 0.97 |
| $k = 7$ | Q1 | 908 | 0.187 | 0.553 | 0.251 | 0.009 |
| | Q2 | 536 | 0 | 0.099 | 0.754 | 0.147 |
| | Q3 | 354 | 0 | 0.006 | 0.249 | 0.746 |
| | Q4 | 370 | 0 | 0 | 0.016 | 0.984 |

cities in the "H-H" pattern has gradually increased, implying that China's IGGPLs have the potential to increase.

The above analyses of Moran's index show that there are global and local spatial correlations in China's IGGPLs and a spatial pattern of club convergence. To explore the reasons for these spatial patterns, we use the traditional and spatial Markov chains to analyze the state transfer probability of IGGPLs. Based on 25%, 50%, and 75% quartiles, we divide the IGGPLs of all cities into four intervals, i.e., [0, 0.783), [0.783, 0.951), [0.951, 1.191), and [1.191, 5.602], which correspond to the four levels of Q1, Q2, Q3, and Q4, respectively. Without considering the IGGPLs of neighboring cities, we calculate the traditional Markov chain transfer probability matrix (shown in Table 3) by Matlab2022a software, and the time spans are 1, 3, 5, and 7 years during 2006–2020. The elements on the diagonal are the probabilities of maintaining original IGGPLs, and the elements in the upper (below) part of the diagonal are the probabilities of transferring from low IGGPLs (high IGGPLs) to high IGGPLs (low IGGPLs). It can be found that in the short term, IGGPLs remain stable and are influenced by their original IGGPLs, with path-dependent characteristics. As a time span of 1 year for example, the minimum value of the element on the diagonal is 0.768, and the maximum value of the element on the non-diagonal is 0.212, with the former being greater than the latter. This means that the probability that the IGGPL will remain at its original level is greater than the probability that it will transfer, and it is this weaker transition that leads to the "L-L" and "H-H" agglomeration characteristics. The IGGPLs in China have shown a continuously improving trend. This is mainly reflected in the fact that IGGPLs remain stable, and the probabilities of transferring to high IGGPLs (low IGGPLs) have gradually decreased (increased). For example, when the time spans are 1, 3, 5, and 7 years, the probabilities of IGGPLs belonging to Q2 remaining stable are 0.768, 0.469, 0.218, and 0.099, respectively, showing a decreasing trend; the probabilities of IGGPLs belonging to Q2 transferring to Q1 are 0.042, 0.032, 0.020, and 0.006, respectively, also showing a decreasing trend; however, the probabilities of IGGPLs belonging to Q2 transferring to Q3 are 0.212, 0.517, 0.739, and 0.754, respectively, showing an increasing trend. In addition, the IGGPLs in China tend to "leapfrog" as the time span increases. For example, as

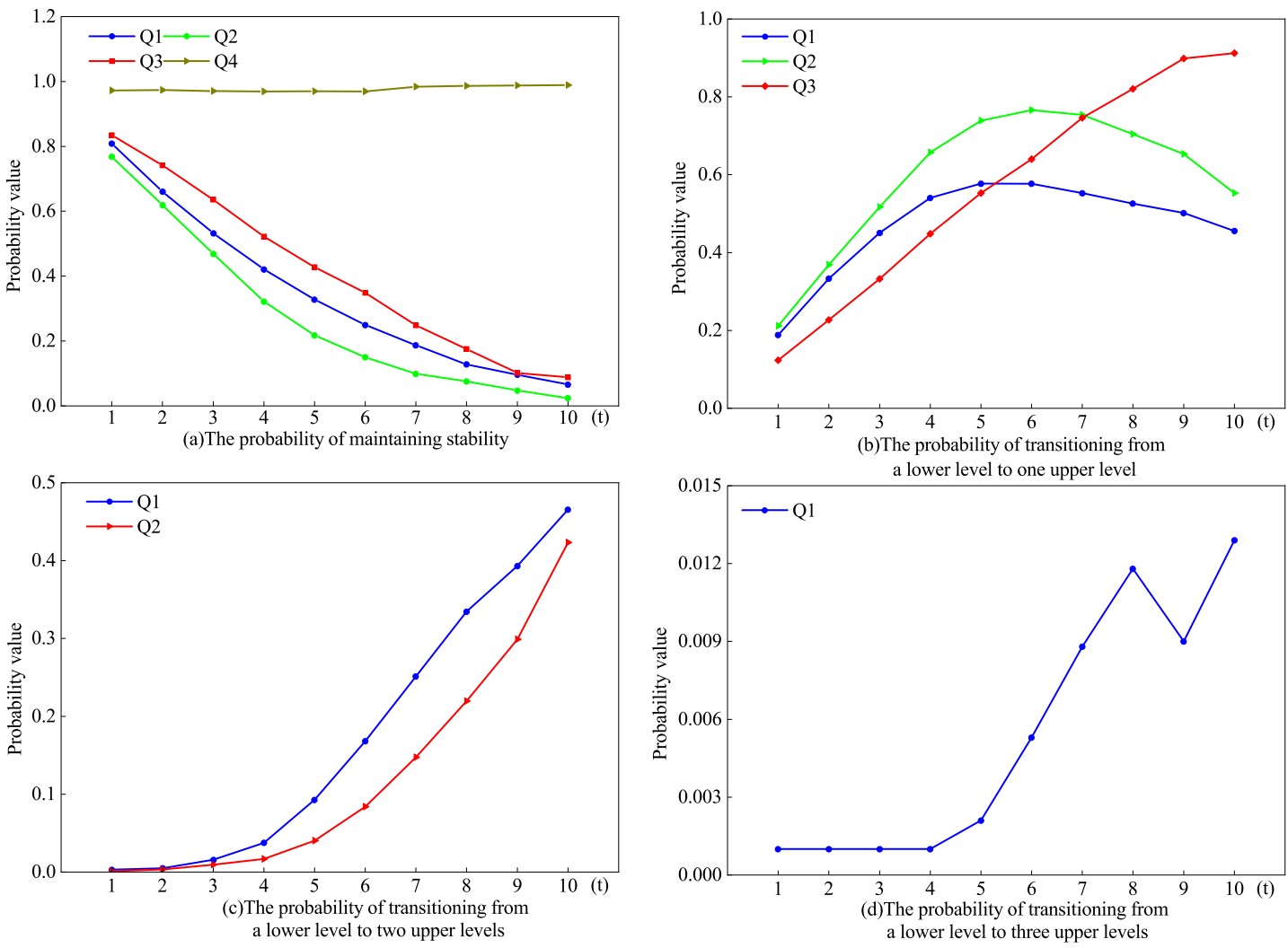

**Fig 7. The probabilities of IGGPLs remaining stable and moving to high-level IGGPLs.**

the time span goes from 1, 3, 5, and 7, the probabilities of IGGPLs belonging to Q1 transferring to Q3 are 0.003, 0.016, 0.093, and 0.251, respectively; the probabilities of IGGPLs belonging to Q2 transferring to Q4 are 0.001, 0.009, 0.040, and 0.147, respectively; and the probabilities of IGGPLs belonging to Q1 transferring to Q4 are 0.001, 0.001, 0.002, and 0.009, respectively, all showing increasing trends. Fig 7 shows in detail the probabilities of IGGPLs remaining stable and transferring to higher IGGPLs, and it can be found that there is a cumulative causal effect on the transfer of IGGPLs.

The traditional Markov chain ignores the influence of the neighboring city's IGGPLs on the state transfers of local IGGPLs. We further construct the spatial Markov chain transfer probability matrix to analyze these (as shown in Table 4). It can be noticed that the state transfers of local IGGPLs are indeed affected by neighboring cities. When the time span is the same if the IGGPLs of neighboring cities are higher than the local IGGPL, then neighboring city IGGPLs have a "pull-up" effect on local IGGPL. For example, when the time span is 1 year, for a city with IGGPL at Q1, the probability of maintaining Q1 when the IGGPLs of its neighboring

**Table 4. Transfer probability matrix of spatial Markov chain.**

| Neighborhood level | Local level | $t = 1$ | | | | | | | | | | $t = 3$ | | | | |
|---|---|---|---|---|---|---|---|---|---|---|---|
| | | $n$ | Q1 | Q2 | Q3 | Q4 | $n$ | Q1 | Q2 | Q3 | Q4 |
| Q1 | Q1 | 592 | 0.902 | 0.095 | 0.003 | 0 | 592 | 0.684 | 0.309 | 0.007 | 0 |
| | Q2 | 108 | 0.009 | 0.824 | 0.167 | 0 | 107 | 0 | 0.43 | 0.561 | 0.009 |
| | Q3 | 75 | 0 | 0.067 | 0.8 | 0.133 | 75 | 0 | 0.08 | 0.493 | 0.427 |
| | Q4 | 28 | 0 | 0 | 0 | 1 | 28 | 0 | 0 | 0 | 1 |
| Q2 | Q1 | 324 | 0.707 | 0.287 | 0.003 | 0.003 | 319 | 0.351 | 0.621 | 0.025 | 0.003 |
| | Q2 | 384 | 0.036 | 0.805 | 0.159 | 0 | 361 | 0.011 | 0.521 | 0.46 | 0.008 |
| | Q3 | 166 | 0 | 0.036 | 0.831 | 0.133 | 147 | 0 | 0.027 | 0.626 | 0.347 |
| | Q4 | 117 | 0.009 | 0.009 | 0.017 | 0.966 | 101 | 0 | 0.02 | 0 | 0.98 |
| Q3 | Q1 | 94 | 0.596 | 0.404 | 0 | 0 | 90 | 0.211 | 0.744 | 0.044 | 0 |
| | Q2 | 390 | 0.01 | 0.736 | 0.251 | 0.003 | 302 | 0 | 0.46 | 0.53 | 0.01 |
| | Q3 | 409 | 0 | 0.046 | 0.856 | 0.098 | 296 | 0 | 0.03 | 0.726 | 0.243 |
| | Q4 | 216 | 0 | 0.005 | 0.051 | 0.944 | 172 | 0 | 0.006 | 0.058 | 0.936 |
| Q4 | Q1 | 6 | 0.333 | 0.667 | 0 | 0 | 6 | 0 | 1 | 0 | 0 |
| | Q2 | 91 | 0 | 0.681 | 0.319 | 0 | 77 | 0 | 0.312 | 0.675 | 0.013 |
| | Q3 | 250 | 0 | 0.032 | 0.812 | 0.156 | 171 | 0 | 0.018 | 0.55 | 0.433 |
| | Q4 | 544 | 0 | 0 | 0.017 | 0.983 | 408 | 0 | 0 | 0.02 | 0.98 |
| Neighborhood level | Local level | $t = 5$ | | | | | | | | | | $t = 7$ | | | | |
| | | $n$ | Q1 | Q2 | Q3 | Q4 | $n$ | Q1 | Q2 | Q3 | Q4 |
| Q1 | Q1 | 592 | 0.459 | 0.465 | 0.076 | 0 | 581 | 0.27 | 0.532 | 0.191 | 0.007 |
| | Q2 | 105 | 0 | 0.143 | 0.79 | 0.067 | 102 | 0 | 0.049 | 0.775 | 0.176 |
| | Q3 | 75 | 0 | 0.067 | 0.227 | 0.707 | 73 | 0 | 0.027 | 0.123 | 0.849 |
| | Q4 | 28 | 0 | 0 | 0 | 1 | 27 | 0 | 0 | 0 | 1 |
| Q2 | Q1 | 295 | 0.149 | 0.746 | 0.098 | 0.007 | 263 | 0.049 | 0.627 | 0.316 | 0.008 |
| | Q2 | 308 | 0.006 | 0.231 | 0.724 | 0.039 | 240 | 0 | 0.1 | 0.817 | 0.083 |
| | Q3 | 118 | 0 | 0.025 | 0.432 | 0.542 | 88 | 0 | 0 | 0.25 | 0.75 |
| | Q4 | 77 | 0 | 0.026 | 0.013 | 0.961 | 53 | 0 | 0 | 0.075 | 0.925 |
| Q3 | Q1 | 77 | 0.026 | 0.818 | 0.156 | 0 | 59 | 0 | 0.475 | 0.492 | 0.034 |
| | Q2 | 221 | 0 | 0.258 | 0.729 | 0.014 | 152 | 0 | 0.151 | 0.664 | 0.184 |
| | Q3 | 203 | 0 | 0.01 | 0.547 | 0.443 | 119 | 0 | 0 | 0.378 | 0.622 |
| | Q4 | 136 | 0 | 0.007 | 0.044 | 0.949 | 104 | 0 | 0 | 0.019 | 0.981 |
| Q4 | Q1 | 6 | 0 | 0.333 | 0.667 | 0 | 5 | 0 | 0 | 1 | 0 |
| | Q2 | 59 | 0 | 0.136 | 0.763 | 0.102 | 42 | 0 | 0.024 | 0.667 | 0.31 |
| | Q3 | 116 | 0 | 0 | 0.345 | 0.655 | 74 | 0 | 0 | 0.162 | 0.838 |
| | Q4 | 294 | 0 | 0 | 0.02 | 0.98 | 186 | 0 | 0 | 0 | 1 |

cities are Q2, Q3, and Q4 are 0.707, 0.596, and 0.333, respectively, which are smaller than 0.808 when the neighboring cities are not considered; for a city with IGGPL at Q2, the probability of maintaining Q2 when the IGGPLs of its neighboring cities are Q3, and Q4 are 0.736 and 0.681, respectively, which are smaller than 0.768 when the neighboring cities are not considered; for a city with IGGPL at Q3, the probability of maintaining Q3 when the IGGPLs of its neighboring cities are Q4 is 0.812, which is smaller than 0.834 when the neighboring cities are not considered. In addition, for cities with the same neighborhoods, the probabilities of IGGPLs remaining at their original level or transferring to lower IGGPLs are decreasing, but the probabilities of transferring to higher IGGPLs are increasing with the increases in the spans. For example, for a city with IGGPL at Q1, when its neighboring cities have IGGPLs at

Q1, the probabilities of remaining at Q1 are 0.902, 0.684, 0.459, and 0.270, respectively, with a decreasing trend as the time span is extended from 1 year to 3, 5, and 7 years, but the probabilities of transferring to higher IGGPLs are increasing gradually. Overall, there are global and local spatial correlations of IGGPLs in China, forming the spatial characteristics of "L-L" and "H-H" agglomerations. The changes in IGGPLs in each city are path-dependent, showing the effect of cumulative causality. In addition, the IGGPLs of a city will be influenced by the IGGPLs of its neighboring cities, showing a "neighbor-high enhancement" effect.

## Conclusions and discussions

### Conclusions

This article proposes the concept of inclusive green growth, establishes an evaluation index system for IGGPL, and calculates the IGGPLs in 271 cities across China from 2006 to 2020 based on the index system. We analyze the temporal and spatial evolutions, regional disparities, and spatial correlations of China's IGGPLs. The findings of this article can provide insights for the effectiveness evaluation of China's inclusive green growth strategy and related research on sustainable development. The main conclusions are as follows. Firstly, China's IGGPLs continue to improve. The eastern IGGPL is far ahead and affects China's IGGPL more strongly. The IGGPL in the northeast region is only second to that in the east region, and after the high-speed growth, it gradually enters a low-speed growth. The IGGPLs in the western and central regions are on par, but the development potential of IGGPLs in the western region is higher than that in the central region. In addition, among the three dimensions, the environment dimension has the highest index, followed by the social inclusion dimension, and then the economy dimension. Secondly, the distributions and dynamic evolutions of IGGPLs in the four regions are different, with the development trend of IGGPLs in the eastern and central regions better. The growth rates of IGGPLs in the western and northeastern regions are slower than those in the eastern and central regions, and there are certain polarizations in the IGGPLs within the two regions, but these polarizations have significantly weakened over time. Thirdly, there are regional differences in China's IGGPL, but these differences gradually decrease over time. In terms of the sources and contributions of differences, regional differences are the main source of total differences. Fourthly, the IGGPL of each city in China has spatial correlation, forming a spatial morphology characterized by "L-L" and "H-H" agglomerations. The changes in IGGPL in various cities exhibit a cumulative causal effect. The state transfer of IGGPL of a city is affected by its neighborhood cities. The cities with high IGGPLs have a "spatial spillover" effect on their neighboring cities with low IGGPLs, which is the main reason for spatial agglomeration.

### Deficiencies and implications for future research

This study has certain limitations. We refer to a large body of related research and follow strict procedures in constructing the index system of IGGPL, and thus measure the IGGPL in China. However, due to data limitations, some indicators are not included in this index system. There is room for improving the index system of IGGPL. The research method and ideas of this paper have a certain universality, and the following aspects can be further studied in the future. Firstly, we can take the county-level IGGPL in China as the research subject, explore the evolutionary characteristics of the county-level IGGPLs, and thus establish the evaluation index systems of IGGPLs in China across "national—economic belt—province—city—county" levels. In addition, we can compare the IGGPLs of major cities in China with those in other countries by constructing an index system capable of international comparisons.

## Supporting information

**S1 Appendix.**
(DOCX)

## Acknowledgments

We are thankful to the editor and anonymous reviewers for meticulously going through the manuscript and for their great suggestions.

## Author Contributions

**Data curation:** Lu Li.

**Funding acquisition:** Lu Li, Shujian Xiang.

**Methodology:** Yingchao Xu.

**Supervision:** Shujian Xiang.

**Writing – original draft:** Yingchao Xu.

**Writing – review & editing:** Lu Li.

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
