## [Decision Letter · Decision Letter 0]

23 Feb 2024

PONE-D-24-03913Performance Evaluation of Inclusive Green Growth in China: Dynamic Evolution, Regional Differences, and Spatial CorrelationPLOS ONE

Dear Dr. Li, 

Thank you for submitting your manuscript to PLOS ONE. After careful consideration, we feel that it has merit but does not fully meet PLOS ONE’s publication criteria as it currently stands. Therefore, we invite you to submit a revised version of the manuscript that addresses the points raised during the review process.

We look forward to receiving your revised manuscript.

Kind regards,

Qingsong He, Ph.D.

Academic Editor

PLOS ONE

Journal Requirements:

 a) The name of the colleague or the details of the professional service that edited your manuscript. 

 b) A copy of your manuscript showing your changes by either highlighting them or using track changes (uploaded as a *supporting information* file).

 c) A clean copy of the edited manuscript (uploaded as the new *manuscript* file).

5. Please be informed that funding information should not appear in the Acknowledgments section or other areas of your manuscript. We will only publish funding information present in the Funding Statement section of the online submission form. Please remove any funding-related text from the manuscript. 

6. We note that Figure 1 in your submission contain map images which may be copyrighted. All PLOS content is published under the Creative Commons Attribution License (CC BY 4.0), which means that the manuscript, images, and Supporting Information files will be freely available online, and any third party is permitted to access, download, copy, distribute, and use these materials in any way, even commercially, with proper attribution. For these reasons, we cannot publish previously copyrighted maps or satellite images created using proprietary data, such as Google software (Google Maps, Street View, and Earth). For more information, see our copyright guidelines: http://journals.plos.org/plosone/s/licenses-and-copyright.

(1) You may seek permission from the original copyright holder of Figure 1 to publish the content specifically under the CC BY 4.0 license.  

Reviewers' comments:

Reviewer's Responses to Questions

**Comments to the Author**

1. Is the manuscript technically sound, and do the data support the conclusions?

Reviewer #1: Partly

Reviewer #2: Yes

Reviewer #3: Yes

Reviewer #4: Yes

2. Has the statistical analysis been performed appropriately and rigorously? 

Reviewer #1: No

Reviewer #2: Yes

Reviewer #3: Yes

Reviewer #4: Yes

3. Have the authors made all data underlying the findings in their manuscript fully available?

Reviewer #1: No

Reviewer #2: Yes

Reviewer #3: No

Reviewer #4: Yes

4. Is the manuscript presented in an intelligible fashion and written in standard English?

Reviewer #1: Yes

Reviewer #2: Yes

Reviewer #3: Yes

Reviewer #4: No

5. Review Comments to the Author

Reviewer #1: The reviewer believes that the topic “Performance Evaluation of Inclusive Green Growth in China: Dynamic Evolution, Regional Differences, and Spatial Correlation” is worthy of investigation. However, the following needs to be addressed. There are minor and major issues that should be corrected. I believe the paper could be further strengthened by added information about.

Please reorganize the manuscript at the journal request. Please change the reference format.

The language of this manuscript is very bad and needs help from native speakers.

The title of the manuscript should fully demonstrate the content of this study and the relevant subjects.

Abstracts should include the purpose and findings of the study.

Introduction . This a very vague statement. These sentences do not provide any information on how the concept could be conceptualized?

This section should explain the study's context and research objective. Furthermore, the research gap needs to be narrowed after analyzing the previous studies. The research method is not adequately explained in the first section.

-Introduction, what authors wanted to convey. Here author must build research gap following the previous studies.-The manuscript does not answer the following concerns: Why is it timeliness to explore such a study? What makes this study different from the previously published studies? Are there any similarly findings in line with the previously published studies? Are the findings different from prior academic studies that were conducted elsewhere, if any? For example, information innovation and innovation network, what it requires, what are the new technologies, some recent issue highlights the importance. See the following: Digital green value co-creation behavior, digital green network embedding and digital green innovation performance: moderating effects of digital green network fragmentation.

Enhancing digital innovation for the sustainable transformation of manufacturing industry: a pressure-state-response system framework to perceptions of digital green innovation and its performance for green and intelligent manufacturing.

Developing a Conceptual Partner Selection Framework: Digital Green Innovation Management of Prefabricated Construction Enterprises for Sustainable Urban Development.

-Methodology: Model.. I suggest authors here build your main heading on Research and data methodology. Clearly explain the model building process, and what previous studies have used similar models (model testing approach).

There is no flow in the text. It partly depends on the lack of proofreading but also on the fact that many statements and claims are made without being followed up by a clear and logical discussion. It is especially problematic in the Introduction that brings up a number of findings from different areas without linking them together.

Please make sure your conclusions' section underscores the scientific value-added of your paper, and/or the applicability of your findings/results. Highlight the novelty of your study.

In addition to summarizing the actions taken and results, please strengthen the explanation of their significance. It is recommended to use quantitative reasoning comparing with appropriate benchmarks, especially those stemming from previous work. See the following:Developing a Conceptual Partner Matching Framework for Digital Green Innovation of Agricultural High-End Equipment Manufacturing System Toward Agriculture 5.0: A Novel Niche Field Model Combined With Fuzzy VIKOR

More importantly, the choice of the variables should be explained in light of the theory and the prior literature on the topic. The arguments are simply relationships and causes very close to the replication of many studies dealing with the same thing.

The authors should emphasize the important role of digital technology in industrial structure upgrading in future research. Some recent issue highlights the importance: The Interaction Mechanism and Dynamic Evolution of Digital Green Innovation in the Integrated Green Building Supply Chain.

Please consider this structure for manuscript final part.

-Discussion

-Conclusion

-Managerial Implication

-Practical/Social Implications

-Discussion needs to be a coherent and cohesive set of arguments that take us beyond this study in particular, and help us see the relevance of what authors have proposed. Authors should create an independent “Discussion” section. Author need to contextualize the findings in the literature, and need to be explicit about the added value of your study towards that literature. Also other studies should be cited to increase the theoretical background of each of the method used. Findings should be contextualized in the literature and should be explicit about the added value of the study towards the literature (New Energy-Driven Construction Industry: Digital Green Innovation Investment Project Selection of Photovoltaic Building Materials Enterprises Using an Integrated Fuzzy Decision Approach). Limitations and future research.

As any emprical study that use different approaches I would like to ask to introduce in the Conclusion section at least a paragraph containing the study limitations. I noticed some things in the paper but a synthesis of statements related to how the study is useful (or partially useful, since are required certain further analysis) and helps potential interested readers does not really exist. Maybe in addition to the last section of Conclusion it is beneficial to introduce a section called: Discussion.

Reviewer #2: Inclusive green growth is an essential approach to achieving sustainable development. This manuscript constructs a system of IGGPLs indicators from using city panel data, further using the methods of vertical and horizontal scatter degree and linear synthesis. Spatial correlation models, regional difference decomposition models, and spatial state transition models are employed to investigate the spatial distribution characteristics and dynamic evolution trends of IGGPLs.

Of particular interest in this manuscript is the construction of an evaluation indicator system for IGGPLs from the city level, and the measurement and evaluation of IGGPLs in China's 271 cities, which is a relatively novel perspective, as most studies have focused on the macro-administrative scale of provinces. In contrast, further refinement of the evaluation index system can provide specific guidance for the green transformation of the economy.

But before accepting publication, there are still some elements that need further explanation， which will be more conducive to improving the quality of the manuscript:

(1) The introduction section does not adequately clarify the scientific issues. It is not stated why IGGPLs are analysed from a spatial perspective and it is recommended that the authors add this section.

(2) The W matrix used by the authors is the 01 neighbourhood matrix. Would the authors please give full consideration to whether this matrix is sufficiently interpretable and whether a geographic distance matrix, an economic distance matrix, or an economic-geographic nested matrix would be more applicable?

(3) In the results and analyses section, the authors provide a lot of descriptions around the empirical results, but the reasons for the results are not fully elaborated. It is recommended that the authors not only present the empirical results, but also analyse more around the real situation, which may be conducive to a more focused opinion.

In summary, I suggest that the author make minor revisions to the manuscript.

Reviewer #3: Review comments on “Performance Evaluation of Inclusive Green Growth in China: Dynamic Evolution, Regional Differences, and Spatial Correlation”

This research construct an index system for inclusive green growth performance levels (IGGPLs) in Chinese cities.. The paper has some fallouts.

1. In the introduction, the overview of the existing research needs to be improved, and the elaboration of the research necessity is insufficient, and does not introduce the policy background of China's inclusive green growth performance levels (IGGPLs).

2. In section 2.1, this part of literature review only reviews "inclusive growth" and "green growth" respectively, and the authors lack in-depth thinking about the concept of inclusive green growth performance levels (IGGPLs).

3. In section 2.2, this part of literature review contains a lot of content and needs to sort out the logic. It is recommended to add a table to make the content clearer.

4. section 2.3 is not sufficient for the review of existing literature, so it can be considered to be combined into section 2.1 and section 2.2, and the major contributions can be considered to be reflected in the introduction and the theoretical contributions at the end.

5. The theoretical basis for the construction of section 3.1 and section 3.2 index system is insufficient, and the reference of relevant literatures is lacking.

6.section 3.3, "and the data after 2020, due to the pandemic, has stronger fluctuations and more outliers" proposes to delete this sentence, Large fluctuations in the data should not be used as a basis for narrowing the scope of the study.

7. It is suggested that authors refer to PLOS ONE format to further standardize the format of this paper.

In summary, it is recommended a minor revision of the manuscript.

Reviewer #4: 1. The description of the article's research purpose or research contribution/innovation in the introduction is not clear and unambiguous. Why this study is necessary? What policy level problem this study is addressing? How the study is expected to provide any solution to that problem? How does the choice of sample is complementing that problem? Are the results and policies generalizable? The introduction is silent in all these aspects.

2. The article is presented only as an expression of results and lacks in-depth explanation of the reasons behind the results.

3. The literature review is not enough, the contribution made by previous studies has not been clearly expressed, and the author needs to introduce the research methods of previous articles to highlight the innovation of this paper.

4. There are also some grammatical errors formatting errors in the sentence expressions, so please check them carefully.

5. Important parts of the discussion are missing. Authors should focus on describing the differences between the article's research and that of other scholars, thus highlighting the article's relevance and academic value.

6. The article was not written following the correct journal's guidelines to be considered for publication. INTORDCUTION→MRTHOD→RESULTS→DISSCUSION→CONCLUSION.

6. PLOS authors have the option to publish the peer review history of their article (what does this mean?). If published, this will include your full peer review and any attached files.

Reviewer #1: No

Reviewer #2: No

Reviewer #3: No

Reviewer #4: No

---

## [Author Response · Author response to Decision Letter 0]

27 Apr 2024

Dear Editor and Reviewers:

We are glad to receive the feedback of our manuscript entitled “Performance Evaluation of Inclusive Green Growth in China: Dynamic Evolution, Regional Differences, and Spatial Correlation". We want to show our great appreciation to both editor and reviewers for their valuable suggestions and comments. The manuscript has been carefully revised according to reviewer's comments. 

In the Response Letter, the comments are highlighted in blue, followed by point-to-point responses to each one. We sincerely hope the revised manuscript will be accepted and can be published in PLOS ONE. 

Yours sincerely,

Lu Li

---

## [Decision Letter · Decision Letter 1]

29 May 2024

Performance Evaluation of Inclusive Green Growth in China: Dynamic Evolution, Regional Differences, and Spatial Correlation

PONE-D-24-03913R1

Dear Dr. Xu,

We’re pleased to inform you that your manuscript has been judged scientifically suitable for publication and will be formally accepted for publication once it meets all outstanding technical requirements.

Kind regards,

Qingsong He, Ph.D.

Academic Editor

PLOS ONE

Additional Editor Comments (optional):

Reviewers' comments:

Reviewer's Responses to Questions

**Comments to the Author**

1. If the authors have adequately addressed your comments raised in a previous round of review and you feel that this manuscript is now acceptable for publication, you may indicate that here to bypass the “Comments to the Author” section, enter your conflict of interest statement in the “Confidential to Editor” section, and submit your "Accept" recommendation.

Reviewer #3: All comments have been addressed

2. Is the manuscript technically sound, and do the data support the conclusions?

Reviewer #3: Yes

3. Has the statistical analysis been performed appropriately and rigorously? 

Reviewer #3: Yes

4. Have the authors made all data underlying the findings in their manuscript fully available?

Reviewer #3: Yes

5. Is the manuscript presented in an intelligible fashion and written in standard English?

Reviewer #3: Yes

6. Review Comments to the Author

Reviewer #3: (No Response)

7. PLOS authors have the option to publish the peer review history of their article (what does this mean?). If published, this will include your full peer review and any attached files.

Reviewer #3: No

---

## [Editor Report · Acceptance letter]

16 Jul 2024

PONE-D-24-03913R1 

PLOS ONE

Dear Dr. Xu, 

I'm pleased to inform you that your manuscript has been deemed suitable for publication in PLOS ONE. Congratulations! Your manuscript is now being handed over to our production team.

Kind regards, 

on behalf of

Dr. Qingsong He 

Academic Editor

PLOS ONE